# Data-independent Module-aware Pruning for Hierarchical Vision Transformers

**Yang He, Joey Tianyi Zhou**[*]
CFAR, Agency for Science, Technology and Research, Singapore
IHPC, Agency for Science, Technology and Research, Singapore
{He_Yang, Joey_Zhou}@cfar.a-star.edu.sg

## Abstract

Hierarchical vision transformers (ViTs) have two advantages over conventional ViTs. First, hierarchical ViTs achieve linear computational complexity with respect to image size by local self-attention. Second, hierarchical ViTs create hierarchical feature maps by merging image patches in deeper layers for dense prediction. However, existing pruning methods ignore the unique properties of hierarchical ViTs and use the magnitude value as the weight importance. This approach leads to two main drawbacks. First, the "local" attention weights are compared at a "global" level, which may cause some "locally" important weights to be pruned due to their relatively small magnitude "globally". The second issue with magnitude pruning is that it fails to consider the distinct weight distributions of the network, which are essential for extracting coarse to fine-grained features at various hierarchical levels.

To solve the aforementioned issues, we have developed a Data-independent Module-Aware Pruning method (DIMAP) to compress hierarchical ViTs. To ensure that "local" attention weights at different hierarchical levels are compared fairly in terms of their contribution, we treat them as a **module** and examine their contribution by analyzing their information distortion. Furthermore, we introduce a novel weight metric that is solely based on weights and does not require input images, thereby eliminating the **dependence** on the patch merging process. Our method validates its usefulness and strengths on Swin Transformers of different sizes on ImageNet-1k classification. Notably, the top-5 accuracy drop is only 0.07% when we remove 52.5% FLOPs and 52.7% parameters of Swin-B. When we reduce 33.2% FLOPs and 33.2% parameters of Swin-S, we can even achieve a 0.8% higher relative top-5 accuracy than the original model. Code is available at: https://github.com/he-y/Data-independent-Module-Aware-Pruning.

## 1 Introduction

Vision transformers Dosovitskiy et al. (2020); Touvron et al. (2020); Yuan et al. (2021) have achieved state-of-the-art (SOTA) performance in the area of computer vision, including image classification, detection, and segmentation. However, the utilization of self-attention and the removal of the convolutions cause vision transformers Dosovitskiy et al. (2020); Liu et al. (2021) heavy computational burdens and significant parameter counts. Therefore, it is necessary to trim the model to reduce the computational cost and required storage.

It is challenging to apply conventional CNN weight pruning methods directly to Vision Transformers since they have different structures and weight properties Raghu et al. (2021); Park & Kim (2022). The most important structure of modern CNNs such as VGGNet Simonyan & Zisserman (2015) and ResNet He et al. (2016) are convolutional layers. As shown in Figure 1, the convolutional layers of ResNet-50 contribute to 92.0% parameters and 99.9% floating-point operations (FLOPs) of the whole network. In contrast, convolutional layers merely contribute 0.1% to parameters and 0.1% to FLOPs in the Swin Transformer. The special structure of ViTs should thus be considered during the pruning process.

---

[*]Corresponding Author

Although there have been attempts to prune conventional ViTs, the unique characteristics of hierarchical ViTs are not fully explored. Local attention is a fundamental feature of hierarchical ViTs, requiring attention computation within a window rather than over the entire image. While this reduces computation on a large scale, it also poses a question: attention weights within one window cannot be compared with those of another window, even if both windows belong to the same image.

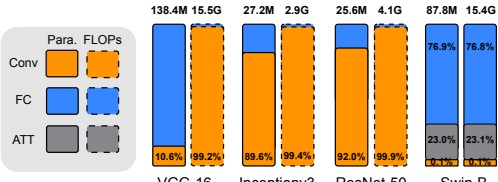

We use an example to explain the problem of the previous magnitude pruning methods. Assume the attention value of window $A$ ranges from 0 to 0.3, and that of window $B$ ranges from 0 to 0.7. Previous magnitude pruning methods Han et al. (2015b) will make this pruning decision: before removing the weights larger than 0.3 in window $B$, all the weights in window $A$ should be removed. How-

Figure 1: Parameters and FLOPs for different components of CNNs (VGG-16, Inceptionv3, ResNet-50) and Swin Transformer. "Conv" = Convolutional layers, "FC" = Fully-connected layers, "ATT" = Attention layers. The numbers on top of the histograms are the total parameters and FLOPs of the network. The percentage numbers listed inside histograms are the contribution from the corresponding layers. Note that norm layers and down-sampling layers are not included for better visualization.

ever, this decision may not always be correct as pruning all weights in a particular window, such as window $A$, could lead to the loss of an important object in the image. Therefore, when determining the least important attention weight in an image, we should not take magnitude into account, but rather rank them based on their contribution.

The second characteristic of hierarchical ViTs is that image patches are merged at higher hierarchical scales. In contrast, conventional ViTs Dosovitskiy et al. (2020); Touvron et al. (2020) have the same number of image patches at different levels. The patch merging characteristic of hierarchical ViTs is designed to extract features ranging from coarse to fine-grained for dense prediction. To achieve this property, the weight distribution of different layers should adapt to the image patches at those layers. To minimize the impact of varying image patch sizes, the weight importance metric should be data-independent. Using the magnitude value as a weight metric is the easiest way to ensure data independence. However, it is not valid when comparing attention values across windows or weight values across layers.

In the paper, we propose a Data-independent Module-Aware Pruning method (DIMAP) to address the above two problems together. First, we include different layers within a module and evaluate weight importance at the level of **module** rather than window or layer. By analyzing the Frobenius distortion Park et al. (2020) incurred by a single weight, we can obtain their weight importance and safely compare the contribution of all weights. This allows us to evaluate the importance of weights within a module, regardless of their location within the windows or layers. Further, we analyze the information distortion at the module level and develop a novel data-independent weight importance metric. The proposed data-independent module-aware weight importance metric in DIMAP has several desirable properties. Firstly, it provides a "global" ranking of the importance of weights at the module level without altering the "local" ranking within a layer, making it a generalization method for layer-wise magnitude pruning. Secondly, it does not use any extra parameters and is purely based on weight values, making it simple and efficient. Finally, it can be applied without the need for complex pruning sensitivity analysis, allowing for efficient one-shot pruning.

Contributions are summarized as follows: (1) We analyze the Frobenius distortion incurred by single-weight and module-level pruning. (2) We propose a data-independent weight importance metric to address the issues of previous pruning methods. (3) Experiments validate that our results are comparable to the state-of-the-art results on image classification benchmarks.

## 2 RELATED WORKS

**CNNs Pruning. 1) Weight Pruning.** Weight pruning LeCun et al. (1990); Han et al. (2015b;a) aims to prune the fine-grained weights of the network. For example, Han et al. (2015b) proposes to remove the small weights whose absolute values are below the threshold. Guo et al. (2016) proposes to discard weights dynamically during the training process. Lebedev & Lempitsky (2016) utilizes the group-sparsity regularization on the loss function to shrink some entire groups of weights toward

zeros. However, these weight pruning methods are designed for CNNs, so it is difficult to directly apply these methods to vision transformers with different properties Raghu et al. (2021); Park & Kim (2022). **2) Filter Pruning.** Filter pruning Li et al. (2017); He et al. (2017); Luo et al. (2017); He et al. (2018a; 2019); Suau et al. (2018); He & Xiao (2023) removes the filters entirely to obtain models with structured sparsity, so the pruned convolutional models can achieve better acceleration. Li et al. (2017) uses L1-norm to evaluate the importance of filters of the network. He et al. (2018b) leverages reinforcement learning to find the redundancy for each layer automatically. However, there are not so many "filters" in vision transformers, so we are not able to utilize these algorithms.

**Variants of Vision Transformer.** Vision transformer is firstly proposed in ViT Dosovitskiy et al. (2020), which utilizes attention not on pixels but instead on small patches of the images. After this, different variants of vision transformers Srinivas et al. (2021); Yuan et al. (2021); Wang et al. (2021); Chen et al. (2021a); Chu et al. (2021b); Han et al. (2021); Xu et al. (2021); Jiang et al. (2021); Xia et al. (2022); Mangalam et al. (2022); Gong et al. (2022); Li et al. (2022b); Mehta & Rastegari (2022); Song et al. (2022); Shao et al. (2022); Tang et al. (2022a;b); Yang et al. (2022a) are emerging. DeiT Touvron et al. (2020) introduces a distillation token to make the student network learns from the teacher network through attention. BoTNet Srinivas et al. (2021) proposes to replace the spatial convolutions with global self-attention in the final three bottleneck blocks of a ResNet. T2T-ViT Yuan et al. (2021) introduces a layer-wise Tokens-To-Token transformation to train the vision transformer from scratch. PVT Wang et al. (2021) utilizes a pyramid structure for dense prediction tasks. CrossViT Chen et al. (2021a) learns multi-scale feature representations with ViT. TNT Han et al. (2021) introduces Transformer in Transformer for excavating features of objects in different scales and locations. CoaT Xu et al. (2021) empowers image transformers with enriched multi-scale and contextual modeling capabilities. LV-ViT Jiang et al. (2021) proposes a new training loss for ViT by taking advantage of all the patch tokens. Swin Transformer Liu et al. (2021) attracts attention for its hierarchical structure and efficient shift window attention.

**Vision Transformer Compression.** Some parallel methods Heo et al. (2021); Graham et al. (2021); Pan et al. (2021); Yue et al. (2021); Shu et al. (2021); Chu et al. (2021a); Yang et al. (2021b); Liang et al. (2022); Yin et al. (2022); Zhang et al. (2022a); Yang et al. (2022b); Li et al. (2022a); Chen et al. (2022b); Zhang et al. (2022b); Yu et al. (2022); Yu & Xiang (2023); Yang et al. (2023); Chuanyang et al. (2022) are proposed for vision transformer compression and acceleration. **1)** This direction focuses on the redundancy of networks, and the structures of the original network are mostly kept. An important direction is to reduce the **input image tokens** Lee et al. (2023). For example, DynamicViT Rao et al. (2021) prunes redundant tokens progressively. EViT Liang et al. (2022) reorganizes the token to reduce the computational cost of multi-head self-attention. SViTE Chen et al. (2021c) proposes a sparse ViT with a trained token selector. **2)** Another direction is to handle the **network itself**. VTP Zhu et al. (2021) prunes the unimportant features of the ViT with sparsity regularization. AutoFormer Chen et al. (2021b) uses an architecture search framework for vision transformer search. As-ViT Chen et al. (2022c) automatically scales up ViTs. UVC Yu et al. (2021) assembles three effective techniques, including pruning, layer skipping, and knowledge distillation, for efficient ViT. NViT Yang et al. (2021a) uses structural pruning with latency-aware regularization on all parameters of the vision transformer. Most previous compression literature Chen et al. (2021c); Zhu et al. (2021); Chen et al. (2022c; 2021b) focus on pruning conventional ViTs such as DeiT Touvron et al. (2020), which use global self-attention. In contrast, hierarchical ViTs such as Swin Transformer utilize local self-attention to save computational costs. Therefore, it is more difficult to compress hierarchical ViTs than conventional ViTs.

## 3 MODULE-AWARE PRUNING

### 3.1 PRELIMINARIES

Assuming a Swin Transformer has $L$ layers, and $\mathbf{W}^{(l)} \in \mathbb{R}^{K \times K \times N_I^{(l)} \times N_O^{(l)}}$ is the weight for the $l_{th}$ layer, where $K$ is the kernel size, $N_I^{(l)}$ and $N_O^{(l)}$ is the number of input and output channels, respectively. The first layer of the Swin Transformer is the patch-embedding layer. It is a convolutional layer, so $K > 1$ and the dimension of this patch-embedding layer is 4. For the latter attention-related layers and fully-connected layers, the dimension of these layers is 2, so we can view the value of $K$ as $K = 1$. LayerNorm (LN) layers are 1-dimension vectors, and these layers only contribute a small portion of FLOPs and storage.

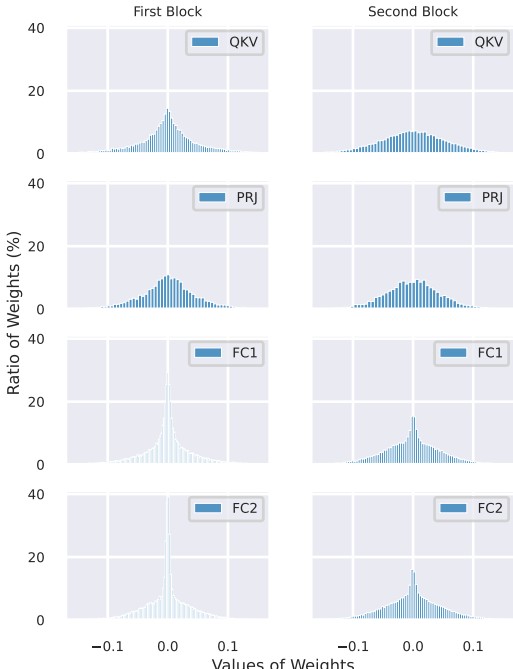

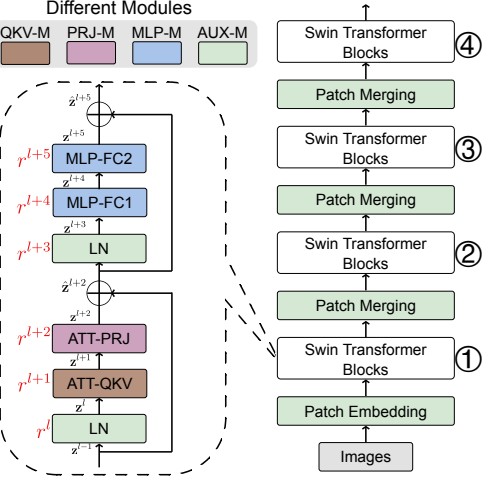

(a) Different modules inside a swin transformer block

(b) Swin Transformer network with four stages: ① ② ③ ④

Figure 3: Categorize network layers to different modules regarding (a) Swin Transformer block; (b) Swin Transformer network. The left figure contains detailed layers of a Swin Transformer block. There are four stages in Swin Transformer. Different colors represent different modules, including the attention-related module (QKV-M and PRJ-M), the multilayer perceptron-related module (MLP-M), and the auxiliary module (AUX-M).

Figure 2: Weight distributions for different functional layers for the first and second Swin Transformer block. The meaning of QKC, PRJ, FC1 and FC2 can be found in Figure 3. The x-axis indicates the values of the weights, and the y-axis denotes the ratios of weights.

**Weight Distributions.** The weight distributions for different functional layers are shown in Figure 2. The first observation is a notable difference in the weight distribution of the QKV layer between the first and second blocks: the QKV layer of the first block contains a significantly higher number of zeros than that of the second block. The dissimilarity in the distributions supports our statement that a direct comparison of attention with their magnitude is not appropriate. Moreover, even with the same block, different functional layers have various distributions. Applying a magnitude pruning threshold of 0.03 would lead to the removal of approximately 30% of the weights from the QKV and PRJ layers, but up to 80% of the weights from the FC1 and FC2 layers. It is clear that magnitude pruning results in unbalanced pruning outcomes for these layers. To ensure a fair comparison of weights across different layers, we employ the information distortion analysis as the weight metric for pruning, as shown in Sec. 3.2.

**Module Definition** The definition of modules is shown in Figure 3. More details can be found in Appendix B.

## 3.2 DISTORTION ANALYSIS AND WEIGHT METRIC

**Single Layer Analysis.** We first utilize a single fully-connected layer [1] to consider the pruning problem from the perspective of minimizing $\ell_2$ distortion after pruning Neyshabur et al. (2015); Lee et al. (2020). Supposing an input $x \in \mathbb{R}^n$ and a weight tensor $W \in \mathbb{R}^{m \times n}$, the pruning operation can be viewed as $\widetilde{W} := M \odot W$, where $M$ is a binary mask whose entries are either 0 or 1. $s$ is the sparsity constraint $\|M\|_0 \leq s$. We aim to find a pruning mask $M$ to minimize the difference between the output from a pruned and an unpruned layer:

$$\min_{\|M\|_0 \leq s} \sup_{\|x\|_2 \leq 1} \|Wx - (M \odot W)x\|_2 \tag{1}$$

---

[1]Attention layers include similar matrix multiplications for Q, K and V.

where $\|x\|_2 \leq 1$ is supported on the unit ball. Then the $\ell_2$ distortion can be bounded as:

$$\|Wx - (M \odot W)x\|_2 \leq \|W - M \odot W\|_2 \cdot \|x\|_2 \tag{2}$$

Therefore, solving Equation 1 is equivalent to solving:

$$\min_{\|M\|_0 \leq s} \|W - M \odot W\| \tag{3}$$

where $\|\cdot\|$ denotes the spectral norm $\|W\| = \sup_{\xi \neq 0} \frac{\|W\xi\|_2}{\|\xi\|_2}$. With Cauchy-Schwarz inequality, the operation can be relaxed to Frobenius distortion minimization:

$$\min_{\|M\|_0 \leq s} \|W - M \odot W\|_F \tag{4}$$

**Module Level Analysis.** Suppose the output of a network $W^{(1:L)} = \left(W^{(1)}, \ldots, W^{(L)}\right)$ given the input $x$ is:

$$f\left(x; W^{(1:L)}\right) = W^{(L)} \sigma\left(\cdots W^{(2)} \sigma\left(W^{(1)}x\right)\cdots\right) \tag{5}$$

As shown in the previous section, a module consists of multiple layers. Suppose $P : \{l_1, l_2, \ldots, l_n\}$ is the set that includes all the layers in the module. Pruning a module aims to minimize the following:

$$\min_{\sum_{l \in P} \|M^{(l)}\|_0 \leq s} \sup_{\|x\|_2 \leq 1} \left\|f\left(x; W^{(1:L)}\right) - f\left(x; \widetilde{W}^{(1:L)}\right)\right\|_2 \tag{6}$$

where $\widetilde{W}^{(1:L)}$ is the network after pruning a module. Since directly solving Equation 6 is difficult, we approximate the distortion in Equation 6 by pruning a single connection. Assume $\widetilde{W}^{(l)} := M^{(l)} \odot W^{(l)}$ indicates the $l_{th}$ weight layer after pruning and $l \in \{l_1, l_2, \ldots, l_n\}$. In this situation, we have an upper bound[2]:

$$\sup_{\|x\|_2 \leq 1} \left\|f\left(x; W^{(1:L)}\right) - f\left(x; W^{(1:l-1)}, \widetilde{W}^{(l)}, W^{(l+1:L)}\right)\right\|_2$$
$$\leq \left\|W^{(l)} - \widetilde{W}^{(l)}\right\|_F \cdot \prod_{\substack{j \neq l \\ j \in [1,L]}} \left\|W^{(j)}\right\|_F \tag{7}$$

**Weight Metric.** With the above analysis of distortion minimization, the last step is to convert the above weight selection process to a data-independent weight importance metric. To rank weights inside a module, we rewrite the right side of the Equation 7 as:

$$\frac{\left\|W^{(l)} - \widetilde{W}^{(l)}\right\|_F}{\left\|W^{(l)}\right\|_F} \cdot \prod_{j \in [1,L]} \left\|W^{(j)}\right\|_F \tag{8}$$

In this formulation, the right side of Equation 8 has no influence on pruning since $\widetilde{W}^{(l)}$ is not included in the product term $\prod_{j=1}^L \left\|W^{(j)}\right\|_F$. Then the distortion of pruning can be evaluated with the left side (the algebraic fraction) of Equation 8. We look into the numerator of the fraction:

$$\left\|W^{(l)} - \widetilde{W}^{(l)}\right\|_F = \sqrt{\sum_{\substack{i \in \{1,\ldots,m\} \\ j \in \{1,\ldots,n\}}} (1 - M_{ij}) W_{ij}^2} \tag{9}$$

where $i$ and $j$ is the index of $W$ and $M$. To minimize Equation 9, for the top largest $W_{ij}^2$, $(1 - M_{ij})$ should be 0.

We sort the weights of $W^{(l)}$ in descending order and get $\{W_{p_1}, W_{p_2}, ..., W_{p_{m \times n}}\}$, where $W_{p_1}$ is the largest weight. In the step of removing $W_{p_j}$, the current $W^{(l)}$ consists of $\{W_{p_1}, W_{p_2}, ..., W_{p_j}\}$

---

[2]Please see proof in Supplementary.

since weights smaller than $W_{p_j}$ are already pruned. After pruning $W_{p_j}$, the current $\widetilde{W}^{(l)}$ consists of $\{W_{p_1}, W_{p_2}, ..., W_{p_{j-1}}\}$. Then we can rewrite the left side of Equation 8 to get the weight importance to minimize the distortion:

$$\text{Imp}(W_{p_j}) := \frac{(W_{p_j})^2}{\sum_{i \le p_j}(W_i)^2} \tag{10}$$

The proposed weight importance has several advantages. 1) This is data-independent and does not require extra parameters other than the weight values. 2) We can conduct one-shot pruning to save computational costs.

## 4 EXPERIMENTS

### 4.1 EXPERIMENTAL SETTING

**Dataset and Architecture.** We follow previous works Chen et al. (2021c); Yu et al. (2021); Chen et al. (2021b) by validating our method on the ImageNet-1K Russakovsky et al. (2015) benchmark dataset. ImageNet-1K contains 1.28 million training images and 50k validation images of $1,000$ classes. In our experiments, the input resolution is $224 \times 224$. The architectures include Swin-T, Swin-S and Swin-B. Note that the numbers of Swin Transformer are cited from the authors' official Github repository[3]. These numbers are slightly different from those reported in their paper Liu et al. (2021): 0.19% higher for Swin-S, 0.14% lower for Swin-T.

**Network Training.** We utilize the same training schedules as Liu et al. (2021) when training. Specifically, we employ an AdamW optimizer for 300 epochs using a cosine decay learning rate scheduler and 20 epochs of linear warm-up. The initial learning rate is 0.001, the weight decay is 0.05, and the batch size is 1024. Our data augmentation strategies are also the same as those of Liu et al. (2021) and include color jitter, AutoAugment Cubuk et al. (2018), random erasing Zhong et al. (2020), mixup Zhang et al. (2017), and CutMix Yun et al. (2019).

**Pruning Setting.** We utilize different pruning ratios to analyze the accuracies for models with different sizes. For the fine-tuning process, we use AdamW optimizer for 30 epochs using a cosine decay learning rate scheduler Loshchilov & Hutter (2017). The base learning rate is 2e-5, and the minimum learning rate is 2e-7. The number of warm-up epochs is 15, and the final learning rate of the linear warm-up process is 2e-8. The weight decay is 1e-8. During pruning, we keep the AUX-M and prune the other three modules.

We follow UVC Yu et al. (2021) in utilizing FLOPs for acceleration measurements and use the number of non-zero parameters to measure the required storage Han et al. (2015b). We compare with state-of-the-art image classifiers variants including ViT Dosovitskiy et al. (2020), DeiT Touvron et al. (2020), BoTNet Srinivas et al. (2021), T2T-ViT Yuan et al. (2021), PVT Wang et al. (2021), CrossViT Chen et al. (2021a), CPVT Chu et al. (2021b), TNT Han et al. (2021), TransMix Chen et al. (2022a), MobileViTv2 Mehta & Rastegari (2023), STViT Chang et al. (2023), Slide-Swin-T Pan et al. (2023) Flatten-Swin-T Han et al. (2023) Long et al. DeiT-S Long et al. (2023) and vision transformer acceleration methods including VTP Zhu et al. (2021), UVC Yu et al. (2021), SViTE Chen et al. (2021c), AutoFormer Chen et al. (2021b), As-ViT Chen et al. (2022c), and AdaViT Chen et al. (2022d), A-ViT Yin et al. (2022), ViT-Slim Chavan et al. (2022), WDPruning Yu et al. (2022), NViT-S Yang et al. (2023), X-Pruner Yu & Xiang (2023). For all of these comparisons, results with the nearest parameters and FLOPs with our methods are listed for a better comparison.

### 4.2 PRUNING SWIN TRANSFORMER

The results of pruning Swin Transformers are shown in Table 1. For every network, we set three target FLOPs reduction ratios, naming them DIMAP1, DIMAP2, DIMAP3. For pruning Swin-B, when we remove 14.3% parameters and 14.4% FLOPs, we can improve top-1 accuracy by 0.04%. This result indicates that pruning Swin Transformer has a similar regularization effect as pruning CNNs.

---

[3]https://github.com/microsoft/Swin-Transformer

| Method | Top-1 acc.(%) | Top-1 acc.↓ (%) | Top-5 acc.(%) | Top-5 acc.↓ (%) | Para. | Para. ↓(%) | FLOPs (%) | FLOPs ↓ (%) |
|---|---|---|---|---|---|---|---|---|
| Swin-B Liu et al. (2021) | 83.48 | – | 96.46 | – | 87.8M | – | 15.4G | – |
| Swin-B-DIMAP1 | **83.52** | **-0.04** | **96.43** | **0.03** | 75.2M | 14.3 | 13.2G | 14.4 |
| Swin-B-DIMAP2 | 83.43 | 0.05 | 96.42 | 0.04 | 58.4M | 33.4 | 10.2G | 33.5 |
| Swin-B-DIMAP3 | 83.28 | 0.20 | 96.39 | 0.07 | **41.7M** | **52.5** | **7.3G** | **52.7** |
| Swin-S Liu et al. (2021) | 83.19 | – | 96.23 | – | 49.6M | – | 8.7G | – |
| Swin-S-DIMAP1 | **83.08** | **0.11** | 96.25 | -0.02 | 42.5M | 14.2 | 7.5G | 14.3 |
| Swin-S-DIMAP2 | 82.99 | 0.20 | **96.26** | **-0.03** | 33.1M | 33.2 | 5.8G | 33.2 |
| Swin-S-DIMAP3 | 82.63 | 0.56 | 96.12 | 0.11 | **23.7M** | **52.2** | **4.1G** | **52.3** |
| Swin-T Liu et al. (2021) | 81.16 | – | 95.48 | – | 28.3M | – | 4.5G | – |
| Swin-T-DIMAP1 | **81.17** | **-0.01** | **95.47** | **0.01** | 24.4M | 13.7 | 3.8G | 13.9 |
| Swin-T-DIMAP2 | 81.11 | 0.05 | 95.42 | 0.06 | 19.2M | 32.0 | 3.0G | 32.4 |
| Swin-T-DIMAP3 | 80.35 | 0.81 | 95.22 | 0.26 | **14.0M** | **50.3** | **2.2G** | **50.8** |

Table 1: Comparison of the pruned Swin Transformer on ImageNet-1K. The "acc. ↓" is the accuracy drop between pruned model and the baseline model (smaller values are better). The negative value of "acc. ↓" means the performance of the pruned model is better than the original model. "Para. ↓" and "FLOPs ↓" are the ratio of removed parameters and FLOPs, respectively.

When pruning about 33.2% of the FLOPs in Swin-S, the top-5 accuracy improves by 0.03%. Although Swin-T is the smallest Swin Transformer model, we can still achieve superior performance. When we remove 13.7% FLOPs or 13.9% FLOPs, the top-1 accuracy drop improves by 0.01%. If we further increase the pruning ratio to 50.8% FLOPs, the top-5 accuracy drop is 0.26%. These results demonstrate that DIMAP works well on variants of the Swin Transformer.

Comparing the "large pruned" to the "small unpruned" model is interesting. We find our pruned models achieve better accuracies with fewer parameters. For example, when we compare Swin-B-DIMAP3 ("large pruned") and Swin-S ("small unpruned"), our Swin-B-DIMAP3 model achieves 0.09% higher accuracy with 7.9M fewer parameters. Similarly, if we look further at Swin-S-DIMAP3 ("large pruned") and Swin-T ("small unpruned"), we find this scenario happens again: our Swin-S-DIMAP3 achieves 0.53% higher accuracy with 4.6M fewer parameters. These results indicate that our DIMAP method effectively shrinks large models into small models and finds models with better accuracy-computation trade-offs than the originally designed models.

### 4.3 COMPARISON WITH STATE-OF-THE-ART METHODS

**Comparison with Vision Transformer Variants (Table 2a).** Our Swin-T-DIMAP2, Swin-S-DIMAP1 and Swin-B-DIMAP3 perform better than PVT-Small, PVT-Medium, PVT-Large Wang et al. (2021), respectively. Our Swin-S-DIMAP3 achieves a 0.33% higher top-1 accuracy with 0.5G fewer FLOPs required than Slide-Swin-T Pan et al. (2023). Our Swin-S-DIMAP3 has a similar improvement over Flatten-Swin-T Han et al. (2023). The Flatten Han et al. (2023) and Slide Pan et al. (2023) methods demonstrate a better trade-off on Swin-S and Swin-B architectures as they focus on improving the attention process. Our proposed method focuses on weight and can be combined with their approaches to achieve higher performance. We only report the results of BAT Long et al. (2023) on DeiT since the Swin transformer is not included in the original BAT paper. These results validate the effectiveness of our DIMAP method.

**Comparison with Vision Transformer Compression Methods (Table 2b).** Our Swin-B-DIMAP3 utilizes 6.3M fewer parameters and 2.7G fewer FLOPs than VTP Zhu et al. (2021) with 2.58% better accuracy. We also achieve 0.7G fewer FLOPs than UVC Yu et al. (2021) with 2.68% higher accuracy. Our Swin-B-DIMAP3 achieves 1.72% better accuracy than SViTE Chen et al. (2021c) with 10.3M fewer parameters and 3.5G fewer FLOPs. Our Swin-S-DIMAP3, Swin-B-DIMAP3, Swin-B-DIMAP1 also performs better than As-ViT-Small Chen et al. (2022c), As-ViT-Base Chen et al. (2022c), As-ViT-Large Chen et al. (2022c), respectively. Compared to X-Pruner Yu & Xiang (2023), our Swin-S-DIMAP2 has 0.99% better accuracy and 0.2G fewer FLOPs. These results demonstrate that our DIMAP achieves a better trade-off between accuracy and computation than existing compression methods.

| method | Image size | Para. | FLOPs | ImageNet top-1 acc. |
|---|---|---|---|---|
| ViT-B/16 Dosovitskiy et al. (2020) | $384^2$ | 86M | 55.4G | 77.9 |
| ViT-L/16 Dosovitskiy et al. (2020) | $384^2$ | 307M | 190.7G | 76.5 |
| DeiT-S Touvron et al. (2020) | $224^2$ | 22M | 4.6G | 79.8 |
| DeiT-B Touvron et al. (2020) | $224^2$ | 86M | 17.5G | 81.8 |
| DeiT-B Touvron et al. (2020) | $384^2$ | 86M | 55.4G | 83.1 |
| BoT-S1-50 Srinivas et al. (2021) | $224^2$ | 20.8M | 4.27G | 79.1 |
| T2T-ViT-t-19 Yuan et al. (2021) | $224^2$ | 39.2M | 9.8G | 82.4 |
| T2T-ViT-t-24 Yuan et al. (2021) | $224^2$ | 64.1M | 15.0G | 82.6 |
| PVT-Small Wang et al. (2021) | $224^2$ | 24.5M | 3.8G | 79.8 |
| PVT-Medium Wang et al. (2021) | $224^2$ | 44.2M | 6.7G | 81.2 |
| PVT-Large Wang et al. (2021) | $224^2$ | 61.4M | 9.8G | 81.7 |
| CrossViT-18† Chen et al. (2021a) | $224^2$ | 44.3M | 9.5G | 82.8 |
| CPVT-S Chu et al. (2021b) | $224^2$ | 22M | – | 79.9 |
| CPVT-B Chu et al. (2021b) | $224^2$ | 86M | – | 81.9 |
| As-ViT-Small Chen et al. (2022c) | $224^2$ | 29.0M | 5.3G | 81.2 |
| As-ViT-Base Chen et al. (2022c) | $224^2$ | 52.6M | 8.9G | 82.5 |
| As-ViT-Large Chen et al. (2022c) | $224^2$ | 88.1M | 22.6G | 83.5 |
| TNT Han et al. (2021) | $224^2$ | 65.6M | 14.1G | 82.9 |
| TransMix Chen et al. (2022a) | $224^2$ | 86.6M | 17.6G | 81.8 |
| MobileViTv2 Mehta & Rastegari (2023) | $224^2$ | 28.3M | 4.5G | 81.3 |
| STViT-Swin-T Chang et al. (2023) | - | - | 3.14G | 81.0 |
| STViT-Swin-S Chang et al. (2023) | - | - | 5.95G | 82.8 |
| STViT-Swin-B Chang et al. (2023) | - | - | 10.48G | 83.2 |
| Slide-Swin-T Pan et al. (2023) | - | 30M | 4.6G | 82.3 |
| Flatten-Swin-T Han et al. (2023) | $224^2$ | 29M | 4.5G | 82.1 |
| BAT-DeiT-S Long et al. (2023) | - | 22.1M | 3.0G | 79.6 |

(a) Vision transformer variants.

| method | image size | #param. | FLOPs | ImageNet top-1 acc. |
|---|---|---|---|---|
| VTP Zhu et al. (2021) | $224^2$ | 48.0M | 10.0G | 80.7 |
| VTP Zhu et al. (2021) | $224^2$ | 67.3M | 13.8G | 81.3 |
| UVC Yu et al. (2021) | $224^2$ | – | 8.0G | 80.6 |
| SViTE Chen et al. (2021c) | $224^2$ | 13.3M | 2.9G | 80.26 |
| $S^2$ViTE Chen et al. (2021c) | $224^2$ | 14.6M | 3.1G | 79.22 |
| SViTE Chen et al. (2021c) | $224^2$ | 52.0M | 10.8G | 81.56 |
| $S^2$ViTE Chen et al. (2021c) | $224^2$ | 56.8M | 11.7G | 82.22 |
| AdaViT Meng et al. (2022) | $224^2$ | – | 3.9G | 81.1 |
| ViT-Slim Chavan et al. (2022) | $224^2$ | 17.7M | 3.7G | 80.6 |
| AutoFormer Chen et al. (2021b) | $224^2$ | 54M | 11G | 82.4 |
| A-ViT Yin et al. (2022) | $224^2$ | 22M | 3.6G | 80.7 |
| WDPruning Yu et al. (2022) | $224^2$ | - | 7.6G | 82.41 |
| WDPruning Yu et al. (2022) | $224^2$ | - | 6.3G | 81.80 |
| NViT-S Yang et al. (2023) | $224^2$ | 21M | 4.2G | 82.19 |
| X-Pruner Yu & Xiang (2023) | $224^2$ | - | 3.2G | 80.7 |
| X-Pruner Yu & Xiang (2023) | $224^2$ | - | 6.0G | 82.0 |
| Swin-T-DIMAP3 | $224^2$ | 14.0M | 2.2G | 80.35 |
| Swin-T-DIMAP2 | $224^2$ | 19.2M | 3.0G | 81.11 |
| Swin-T-DIMAP1 | $224^2$ | 24.4M | 3.8G | 81.17 |
| Swin-S-DIMAP3 | $224^2$ | 23.7M | 4.1G | 82.63 |
| Swin-S-DIMAP2 | $224^2$ | 33.1M | 5.8G | 82.99 |
| Swin-S-DIMAP1 | $224^2$ | 42.5M | 7.5G | 83.08 |
| Swin-B-DIMAP3 | $224^2$ | 41.7M | 7.3G | 83.28 |
| Swin-B-DIMAP2 | $224^2$ | 58.4M | 10.2G | 83.43 |
| Swin-B-DIMAP1 | $224^2$ | 75.2M | 13.2G | **83.52** |

(b) Vision transformer compression methods.

Table 2: Compare results on ImageNet-1K classification.

| Method Ratio | Swin-B 19% | Swin-S 19% | Swin-T 19% | Swin-B 28% | Swin-S 28% | Swin-T 28% | Swin-B 38% | Swin-S 38% | Swin-T 38% | Swin-B 50% | Swin-S 50% | Swin-T 50% |
|---|---|---|---|---|---|---|---|---|---|---|---|---|
| Uniform | 83.25 | 82.91 | 80.70 | 82.27 | 81.91 | 79.20 | 77.73 | 77.86 | 72.06 | 15.26 | 13.93 | 6.02 |
| Ours | 83.36(+0.11) | 82.98(+0.07) | 80.87(+0.17) | 83.02(+0.75) | 82.58(+0.67) | 80.24(+1.04) | 81.95(+4.22) | 81.56(+3.70) | 78.54(+4.79) | 78.13(+62.87) | 76.78(+62.85) | 71.36(+65.34) |

Table 3: Comparison of uniform magnitude-based pruning with our DIMAP. The ratio here means the proportion of removed parameters after pruning. The numbers in brackets are our top-1 accuracy gain over uniform pruning.

## 4.4 ADDITIONAL ANALYSIS

**Module Pruning Rate Analysis.** As shown in Figure 5, we analyze pruning results based on our proposed module-level weight importance. Specifically, we first calculate the module-level weight importance for each module. Then we conduct magnitude pruning to remove the least important weights from the module. Note that a module merely has one threshold for pruning. The pruning thresholds for QKV-M, PRJ-M and MLP-M are 3.8E-7, 4.99E-7 and 1.5E-6, respectively. After pruning, the ratios of the kept parameters for different layers are shown as the y-axis in Figure 5. We have several observations. 1) The lower layers keep a larger ratio of parameters than the higher layers. This phenomenon indicates the higher layers have greater redundancy. 2) In the first and the second Swin Transformer block, we find that the attention-related layers (layer 1,2,5,6) have larger ratios than MLP-related layers (layer 3,4,7,8). This result shows that the attention operation is extremely important for low-level features. 3) We can take a look at Figure 5 and Figure 2 together. Layer 4 and layer 8 in Figure 5 correspond to two FC2 layers in Figure 2. In Figure 5, layer 4 has a smaller weight-kept ratio than layer 8. This pruning result reasonably corresponds to distributions in Figure 2: there are more weights close to zero in layer 4 than in layer 8.

| Model | Multiple Modules (%) | Single Module (%) | Difference (%) |
|---|---|---|---|
| Swin-B-DIMAP3 | 83.28 | 83.17 | 0.11 |
| Swin-S-DIMAP3 | 82.63 | 82.31 | 0.32 |
| Swin-T-DIMAP3 | 80.35 | 80.12 | 0.23 |

Figure 4: Comparison of the theoretical and realistic acceleration. Only the time consumption of the forward procedure is considered.

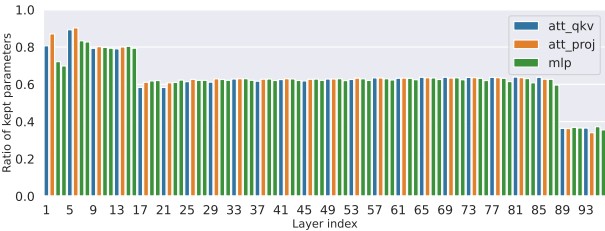

Figure 5: Results of pruning 45% of the parameters from the Swin-S with our module-level weight importance. The x-axis denotes the layer index, and the y-axis indicates the ratio of the remaining parameters. Different colors represent different modules.



Figure 6: Visualization of four pruned layers in Swin-B. The pruning ratio is 50%. The x-axis is the input dimension, and the y-axis indicates the output dimension. The red areas indicate kept weights, and the grey areas represent pruned weights.

**Comparison with Uniform Pruning.** In Table 3, we conduct an ablation study by comparing our method to magnitude-based uniform pruning. Apparently, we are better than uniform pruning at all settings. Explicitly, at a small pruning ratio like 19%, our accuracy gain is just about 0.1%. When we increase the pruning ratio to 50%, our DIMAP method achieves a 60+% accuracy gain over uniform pruning. These results validate that the necessity of DIMAP in guaranteeing performance, especially when the pruning ratio is large.

**Visualization.** We visualize the four DIMAP-pruned layers in the first Swin Transformer block in Figure 6. First, for the figure of ATT-QKV, the y-indexes for "Query, Key, and Value" are [0,127], [128,255], [256,383], respectively. We find the "Query" and "Key" matrix of ATT-QKV is denser than the "Value" matrix. This implies that "Query" and "Key" may suffer more from pruning than "Value". Second, although ATT-PRJ, MLP-FC1 and MLP-FC2 serve as fully-connected layers, the pattern of ATT-PRJ is quite different from the other two layers. A possible reason is that the input of ATT-PRJ also includes the relative position bias of attention, and ATT-PRJ needs to project this position information. Third, we can even find several rows of weights are removed entirely. For example, in MLP-FC1, no weights are kept when the y-index equals 228 and 231. This phenomenon may further guide us on how to design the size of MLP. Fourth, we find that the pattern of MLP-FC1 is similar to MLP-FC2. This result is another proof that we should categorize MLP-FC1 and MLP-FC2 into the same module.

**Model as Module.** It is interesting to view the entire model as a single module. In this setting, we keep the auxiliary layers unpruned and view all other layers as a module. This means all the weights are compared based on our novel weight metric. The results shown in Tab. 4 demonstrate our fine-grained module definition in Sec. 3.1 can achieve better performance than "Model as Module".

## 5 CONCLUSION AND FUTURE WORK

We introduce DIMAP to compress hierarchical ViTs while considering two key properties of these models. We present two main contributions: an analysis of information distortion at the module level, and a data-independent weight importance metric based on the distortion analysis. A drawback of weight pruning is that it does not bring realistic acceleration on GPUs, and we will explore this direction in the future. Furthermore, we are interested in the performances on downstream tasks such as object detection and segmentation. We are also interested in the performance of our method on other ViT variants.

## 6 ACKNOWLEDGEMENT

This work was supported in part by A*STAR Career Development Fund (CDF) under C233312004, in part by the National Research Foundation, Singapore, and the Maritime and Port Authority of Singapore / Singapore Maritime Institute under the Maritime Transformation Programme (Maritime AI Research Programme – Grant number SMI-2022-MTP-06).

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

## A   BOUND PROOF

In this section, we provide proof for the upper bound of pruning process:

$$
\begin{aligned}
&\sup_{\|x\|_2 \leq 1} \left\| f\left(x; W^{(1:L)}\right) - f\left(x; W^{(1:l-1)}, \widetilde{W}^{(l)}, W^{(l+1:L)}\right) \right\|_2 \\
&\leq \left\| W^{(l)} - \widetilde{W}^{(l)} \right\|_F \cdot \prod_{\substack{j \neq l \\ j \in [1,L]}} \left\| W^{(j)} \right\|_F
\end{aligned}
\tag{11}
$$

Inspired by Neyshabur et al. (2015); Lee et al. (2020), we can write the network $f\left(x; W^{(1:L)}\right)$ with its layers:

$$
f\left(x; W^{(1:L)}\right) = W_L \sigma \left(W_{L-1} \sigma \left(W_{L-2}\left(\ldots \sigma\left(W_1 x\right)\right)\right)\right)
\tag{12}
$$

Then we can peel the highest layer $W_L$:

$$
\begin{aligned}
&\left\| f\left(x; W^{(1:L)}\right) - f\left(x; W^{(1:l-1)}, \widetilde{W}^{(l)}, W^{(l+1:L)}\right) \right\|_2 \\
&= \left\| W^{(d)} \left(\sigma\left(f\left(x; W^{(1:L-1)}\right)\right) - \sigma\left(f\left(x; W^{(1:l-1)}, \widetilde{W}^{(l)}, W^{(l+1:L-1)}\right)\right)\right) \right\|_2
\end{aligned}
\tag{13}
$$

With Cauchy-Schwarz inequality Steele (2004), we have the upper bound of Equation 13 as:

$$
\left\| W^{(L)} \right\|_F \cdot \left\| \sigma\left(f\left(x; W^{(1:L-1)}\right)\right) - \sigma\left(f\left(x; W^{(1:l-1)}, \widetilde{W}^{(l)}, W^{(l+1:L-1)}\right)\right) \right\|_2
\tag{14}
$$

Suppose $\sigma$ is the ReLU activation, so we use the 1-Lipschitzness of ReLU activation with respect to $\ell_2$ norm for the upper bound of Equation 14 as:

$$
\left\| W^{(L)} \right\|_F \cdot \left\| f\left(x; W^{(1:L-1)}\right) - f\left(x; W^{(1:l-1)}, \widetilde{W}^{(l)}, W^{(l+1:L-1)}\right) \right\|_2
\tag{15}
$$

Then we continue peeling the second highest layer $W_{L-1}$ to lower layers and stop the peeling at the **pruned layer** $\widetilde{W}^{(l)}$. We have this upper bound for Equation 13:

$$
\begin{aligned}
&\left\| f\left(x; W^{(1:L)}\right) - f\left(x; W^{(1:l-1)}, \widetilde{W}^{(l)}, W^{(l+1:L)}\right) \right\|_2 \\
&\leq \left( \prod_{j>l} \left\| W^{(j)} \right\|_F \right) \cdot \left\| f\left(x; W^{(1:l)}\right) - f\left(x; W^{(1:l-1)}, \widetilde{W}^{(l)}\right) \right\|_2
\end{aligned}
\tag{16}
$$

We consider the effect of pruning for the term on the right side and use Cauchy-Schwarz inequality again:

$$
\begin{aligned}
\left\| f\left(x; W^{(1:l)}\right) - f\left(x; W^{(1:l-1)}, \widetilde{W}^{(l)}\right) \right\|_2 &= \left\| \left(W^{(l)} - \widetilde{W}^{(l)}\right) \sigma\left(f\left(x; W^{(1:l-1)}\right)\right) \right\|_2 \\
&\leq \left\| W^{(l)} - \tilde{W}^{(l)} \right\|_F \cdot \left\| \sigma\left(f\left(x; W^{(1:l-1)}\right)\right) \right\|_2
\end{aligned}
\tag{17}
$$

For the activation term, based on $\sigma(\mathbf{0}) = \mathbf{0}$, we have:

$$
\left\| \sigma\left(f\left(x; W^{(1:l-1)}\right)\right) \right\|_2 = \left\| \sigma\left(f\left(x; W^{(1:l-1)}\right)\right) - \sigma(\mathbf{0}) \right\|_2 \leq \left\| f\left(x; W^{(1:l-1)}\right) - \mathbf{0} \right\|_2 = \left\| f\left(x; W^{(1:l-1)}\right) \right\|_2.
\tag{18}
$$

If we continue the peeling process as shown in equation 13, we can achieve equation 11.

## B   MODULE DEFINITION

**Attention-related Module (QKV-M and PRJ-M).** Self-attention is an important operation in Swin Transformer. In every transformer block, there are two layers related to self-attention, that is, ATT-QKV and ATT-PRJ in Figure 3. ATT-QKV means the "Query, Key, and Value" matrix for self-attention, and ATT-PRJ indicates the projection layer for self-attention. Assuming $\mathbf{z}^l$ is the feature map of $l_{th}$ layer, two attention-related layers are:

$$\mathbf{z}^{l+1} = \text{ATT-QKV}\left(\mathbf{z}^l\right), \quad \mathbf{z}^{l+2} = \text{ATT-PRJ}\left(\mathbf{z}^{l+1}\right) \tag{19}$$

where $\mathbf{z}^l$ is the input feature map of ATT-QKV, and $\mathbf{z}^{l+1}$ is the input feature map of ATT-PRJ. After these two layers, there is a residual connection between the output of the ATT-PRJ layer and the feature map before the LN layer:

$$\hat{\mathbf{z}}^{l+2} = \mathbf{z}^{l+2} + \mathbf{z}^{l-1} \tag{20}$$

Assume there are $J$ transformer blocks in the whole network. For the transformer block shown in Figure 3(a), assume the range of the layer index is $[l, \ l + 5]$. For the $j_{th}$ block after this block, the layer index range is $[l + p_j, \ l + p_j + 5]$, where $p_j$ means the number of layers between these two blocks. Then we can define QKV-M and PRJ-M as:

$$\begin{aligned}
\text{QKV-M} &: \left\{\mathbf{W}^{l+1}, ..., \mathbf{W}^{l+p_j+1}, ...\right\} \quad \text{for} \quad j \in [1, J]. \\
\text{PRJ-M} &: \left\{\mathbf{W}^{l+2}, ..., \mathbf{W}^{l+p_j+2}, ...\right\} \quad \text{for} \quad j \in [1, J].
\end{aligned} \tag{21}$$

**Multilayer Perceptron-related Module (MLP-M).** Another important part of the Swin Transformer is the multilayer perceptron. There are two MLP layers in every Swin Transformer block. We name these two layers MLP-FC1 and MLP-FC2:

$$\mathbf{z}^{l+4} = \text{MLP-FC1}\left(\mathbf{z}^{l+3}\right), \quad \mathbf{z}^{l+5} = \text{MLP-FC2}\left(\mathbf{z}^{l+4}\right) \tag{22}$$

where $\mathbf{z}^{l+3}$ is the input feature map of MLP-FC1, and $\mathbf{z}^{l+4}$ is the input feature map of MLP-FC2. Similarly, after two MLP layers, there is a residual connection between the output of the MLP-FC2 layer and the feature map before the LN layer:

$$\hat{\mathbf{z}}^{l+5} = \mathbf{z}^{l+5} + \hat{\mathbf{z}}^{l+2} \tag{23}$$

Considering $J$ blocks in the whole network, and $j \in [1, J]$, we can define the MLP-M as:

$$\text{MLP-M} : \left\{\mathbf{W}^{l+4}, \mathbf{W}^{l+5}, ..., \mathbf{W}^{l+p_j+4}, \mathbf{W}^{l+p_j+5}, ...\right\}. \tag{24}$$

**Auxiliary Module (AUX-M).** The auxiliary module consists of the auxiliary layers inside and outside the Swin Transformer block. Inside the Swin Transformer block, as shown in Figure 3(a), there are two LN layers. Some other auxiliary layers outside the Swin Transformer blocks, including a patch embedding layer and several patch merging layers, are also included in AUX-M. We do not prune these layers due to their important contribution to the network and their relatively small parameter count.

