# OpenReview forum: "Data-independent Module-aware Pruning for Hierarchical Vision Transformers"
_ICLR.cc/2024/Conference — ICLR 2024 poster_

### Official Review · Reviewer_VGj7 · 2023-10-30

**Soundness:** 4 excellent
**Presentation:** 3 good
**Contribution:** 4 excellent
**Rating:** 8
**Confidence:** 4

**Summary:**

The paper introduces the Data-independent Module-Aware Pruning (DIMAP) for hierarchical vision transformers (ViTs). Instead of relying on magnitude values like traditional pruning techniques, DIMAP evaluates "local" attention weights in hierarchical ViTs by their impact on information distortion. By examining weight distribution at different hierarchical levels, DIMAP offers an alternative to magnitude-based methods. Tests on Swin Transformers highlight its efficiency, with a small decrease in top-5 accuracy and notable reductions in FLOPs and parameters.

**Strengths:**

1. The paper presents DIMAP, a unique pruning method for hierarchical ViTs. Instead of the prevalent focus on weight magnitudes, DIMAP evaluates attention weights based on their effect on information distortion, setting it apart from standard methods.

2. The approach is grounded in solid theory, with mathematical models reflecting a comprehensive grasp of the issue. The choice to use distortion analysis as a measure of weight significance stands out.

3. The paper's organization is logical, detailing the obstacles and answers related to pruning hierarchical ViTs. Its equations, coupled with their explanations, offer transparency and ease of understanding for readers.

4. Given the rising importance of hierarchical ViTs in computer vision, the paper's relevance is significant. Efficient pruning is vital for these transformers in limited-resource scenarios. DIMAP, focusing on maintaining crucial attributes while cutting down computations, meets a current community demand.

**Weaknesses:**

Complexity: While the module-aware method proves efficient, it could complicate the pruning process, particularly for extensive models or vast datasets. How much computational burden does this additional step bring?

The threshold determination is at the module level. Could this be determined at the model level instead? What differentiates these two levels of thresholding?

**Questions:**

The computational cost and threshold explained in weakness section.

---

> ### Author Response · Authors · 2023-11-17
>
> Thank you very much for your hign score! Below, we address your concerns regarding the complexity of our method and the choice of threshold determination.
>
> > 1. Complexity: While the module-aware method proves efficient, it could complicate the pruning process, particularly for extensive models or vast datasets. How much computational burden does this additional step bring?
>
> DIMAP's module-aware approach is designed to be efficient and adds minimal computational burden. This efficiency is achieved through the following key aspects:
>
>
> **Data-Independence**: As highlighted in our method section, DIMAP's weight metric is data-independent, eliminating the need for inference on additional images or datasets for pruning. This significantly reduces the computational cost compared to data-dependent methods.
>
> **One-Shot Pruning**: Unlike iterative pruning methods, DIMAP enables one-shot pruning, reducing the computational overhead. This approach is based on the weight importance metric (Imp), derived as follows:
>
> $$
> \operatorname{Imp}\left(W_{p_j}\right):=\frac{\left(W_{p_j}\right)^2}{\sum_{i \leq p_j}\left(W_i\right)^2}
> $$
>
> This metric efficiently evaluates the importance of weights for pruning, without requiring multiple rounds of pruning and retraining.
>
> > 2. The threshold determination is at the module level. Could this be determined at the model level instead? What differentiates these two levels of thresholding?
>
> The decision to implement thresholding at the module level, as opposed to the model level, is informed by the unique structure and requirements of hierarchical ViTs. Module-level thresholding allows for more granular control and respects the intricate design of these models.
> This approach is supported by our distortion analysis, where we minimize the following for a given module in Eq. (6). This formulation ensures that each module's unique contribution to the overall network functionality is considered during pruning, leading to a more effective and efficient pruning process.

---

### Official Review · Reviewer_uDum · 2023-10-31

**Soundness:** 3 good
**Presentation:** 3 good
**Contribution:** 4 excellent
**Rating:** 6
**Confidence:** 5

**Summary:**

Traditional pruning methods ignore the unique properties of hierarchical ViTs by utilizing magnitude as the weight importance, which leads to issues in weight pruning at both local and global scales. DIMAP addresses these challenges by treating attention weights at different hierarchical levels as a "module" and examining their significance by analyzing their information distortion. The proposed method demonstrates superior performance on Swin Transformers when compared with other approaches, especially in terms of accuracy and efficiency in the pruning process.

**Strengths:**

1. The idea is somewhat novel. The unique properties of hierarchical ViTs leads to the proposing of a module-aware approach for pruning hierarchical ViTs.
2. The paper is well-structured, with clear explanations of the challenges with existing pruning methods, the introduction of their novel method, and mathematical derivations supporting their approach.
3. The experiments are good. The top-5 accuracy drop is only 0.07% when we removing 52.5% FLOPs and 52.7% parameters of Swin-B.

**Weaknesses:**

1. Why it's called "data independent" since the weights are updated with training data?
2. A comparative study with existing pruning methods, including their weaknesses when applied to hierarchical ViTs, could offer more insights into the significant advantages of DIMAP.
3. More information on the implementation specifics, hyperparameters used, and other nuances would be helpful for reproducibility and understanding the intricacies of DIMAP.
4. It's not clear that uniform pruning means pruning parameters uniformly or pruning FLOPs uniformly?
5. While the method has been validated on the Swin Transformer architecture on ImageNet-1k, it would be beneficial to see its performance on other Vision Transformer architectures or datasets to establish its versatility.

**Questions:**

See weakness

---

> ### Author Response · Authors · 2023-11-17
> **Response 1**
>
> Thank you for your constructive feedback regarding our work. We appreciate the opportunity to clarify and expand upon the key aspects of our research. Below, we address each of your concerns in detail.
>
> > 1. Why it's called "data independent" since the weights are updated with training data?
>
> While the **initial training** of hierarchical Vision Transformers (ViTs) indeed utilizes training data, DIMAP's unique aspect lies in its **pruning process**, which operates independently of new or additional data. DIMAP evaluates and prunes the network weights based on their contribution to information distortion, a process that does not require further input data. By analyzing the attention weights and their hierarchical structure, DIMAP can efficiently identify and remove less significant weights, thus optimizing the model post-training without relying on additional datasets.
>
> > 2. A comparative study with existing pruning methods, including their weaknesses when applied to hierarchical ViTs, could offer more insights into the significant advantages of DIMAP.
>
> - **Comparison 1**. As detailed in our manuscript under the section CNNs Pruning (referring to works like LeCun et al. 1990, Han et al. 2015, Guo et al. 2016, Lebedev et al. 2016), traditional weight and filter pruning methods are primarily designed for Convolutional Neural Networks (CNNs). These methods focus on pruning fine-grained weights or entire filters based on specific criteria like weight magnitude or group-sparsity regularization. However, their direct applicability to ViTs is limited due to the distinct architectural differences, as discussed in works by Raghu et al. 2021 and Park et al. 2022. Hierarchical ViTs, such as the Swin Transformer introduced by Liu et al. 2021, exhibit unique characteristics like local self-attention mechanisms, which are not adequately addressed by these CNN-centric pruning approaches.
>
> - **Comparison 2**. In the section Vision Transformer Compression of our manuscript, we discuss methods specifically designed for ViT compression (e.g., Heo et al. 2021, Rao et al. 2021, Zhu et al. 2021, Chen et al. 2021). While these methods address ViT structures, they predominantly focus on conventional ViTs employing global self-attention, as seen in works like DynamicViT by Rao et al. 2021 and AutoFormer by Chen et al. 2021. These methods either prune input image tokens or handle network features directly, but they fall short in effectively compressing hierarchical ViTs, which leverage local self-attention for computational efficiency.
>
> - **Advantages Over Existing Methods**. Our DIMAP method is uniquely positioned to address the challenges posed by hierarchical ViTs. Unlike traditional CNN pruning methods, DIMAP is tailored to the specific needs of hierarchical structures, preserving the integrity of local self-attention mechanisms. Compared to existing ViT compression methods, DIMAP's data-independent approach offers a significant advantage, allowing for efficient pruning without the need for additional input data. This distinction is particularly crucial in the context of hierarchical ViTs, where the complexity and nuances of local attention mechanisms demand a more sophisticated pruning approach.

---

> ### Author Response · Authors · 2023-11-17
> **Response 2**
>
> > 3. More information on the implementation specifics, hyperparameters used, and other nuances would be helpful for reproducibility and understanding the intricacies of DIMAP.
>
> To make it clear. We make a table for the implementation details.
>
> #### Network Training
>
> | Parameter            | Detail                                                        |
> |----------------------|---------------------------------------------------------------|
> | Optimizer            | AdamW                                                         |
> | Epochs               | 300 epochs with cosine decay learning rate scheduler, 20 epochs linear warm-up |
> | Initial Learning Rate| 0.001                                                         |
> | Weight Decay         | 0.05                                                          |
> | Batch Size           | 1024                                                          |
> | Data Augmentation    | Color jitter, Autoaugment, Random erasing, Mixup, Cutmix      |
>
> #### Pruning Setting
>
> | Parameter             | Detail                                                      |
> |-----------------------|-------------------------------------------------------------|
> | Pruning Ratios        | Different ratios for models of varying sizes                |
> | Optimizer             | AdamW                                                       |
> | Epochs                | 30 epochs with cosine decay learning rate scheduler        |
> | Base Learning Rate    | 2e-5                                                        |
> | Minimum Learning Rate | 2e-7                                                        |
> | Warm-up Epochs        | 15                                                          |
> | Final Learning Rate   | 2e-8 (after linear warm-up)                                 |
> | Weight Decay          | 1e-8                                                        |
> | Pruning Strategy      | Keep AUX-M, prune other three modules                       |
>
>
> > 4. It's not clear that uniform pruning means pruning parameters uniformly or pruning FLOPs uniformly?
>
> In our manuscript, "uniform pruning" refers to the consistent reduction of parameters across different layers of the network. This approach ensures a balanced reduction in complexity while maintaining the hierarchical structure essential for ViTs.
>
>
> > 5. While the method has been validated on the Swin Transformer architecture on ImageNet-1k, it would be beneficial to see its performance on other Vision Transformer architectures or datasets to establish its versatility.
>
>
> We acknowledge that our current evaluation focuses on the Swin Transformer architecture using the ImageNet-1k dataset. We have explained why we choose Swin Transformer. We are in the process of extending our evaluation to other Vision Transformer architectures and diverse datasets. This expansion will not only establish DIMAP’s versatility but also provide insights into its adaptability across various domains and applications.

---

### Official Review · Reviewer_qfKu · 2023-11-11

**Soundness:** 3 good
**Presentation:** 4 excellent
**Contribution:** 3 good
**Rating:** 6
**Confidence:** 4

**Summary:**

Hierarchical Vision Transformers (ViTs) offer linear computational complexity concerning image size and create hierarchical feature maps. To solve the aforementioned issues, the authors  have developed a Data-independent Module-Aware Pruning method (DIMAP) to compress hierarchical ViTs. To ensure that "local" attention weights at different hierarchical levels are compared fairly in terms of their contribution, they treat them as a module and examine their contribution by analyzing their information distortion. Furthermore, they introduce a novel weight metric that is solely based on weights and does not require input images, thereby eliminating the dependence on the patch merging process. The experiments are fine to me.

**Strengths:**

The authors introduce a new pruning approach called Data-independent Module-Aware Pruning (DIMAP). DIMAP compares "local" attention weights fairly by treating them as a module and examining their contribution based on information distortion. A new weight metric that doesn't need input images is also introduced, making it data-independent. The method has been validated on Swin Transformers, demonstrating its effectiveness and efficiency.

**Weaknesses:**

To me the extensive applications will be beneficial to be new contributions of field.
Could you discuss more the new applications of your method.

**Questions:**

To me the extensive applications will be beneficial to be new contributions of field.
Could you discuss more the new applications of your method.

---

> ### Author Response · Authors · 2023-11-17
> **Response from authors**
>
> Thank you for your insightful comments and the opportunity to discuss the broader applications of our Data-independent Module-Aware Pruning (DIMAP) method for hierarchical Vision Transformers (ViTs). We appreciate your suggestion to elaborate on the potential impact and utility of DIMAP across various domains. Below, we address this aspect in detail, highlighting the general applications, specific use cases, and future directions.
>
> 1. Contextual Introduction
>
> The development of DIMAP represents a significant advancement in the efficient pruning of hierarchical ViTs. Its unique approach—focusing on module-aware pruning and data independence—addresses critical challenges in model efficiency and privacy concerns. We believe that these features not only contribute to the technical robustness of our method but also enhance its applicability in a wide range of real-world scenarios.
>
> 2. General Applications
>
> DIMAP’s applicability extends across various domains where hierarchical ViTs have shown promise. These include, but are not limited to, image classification, object detection, and semantic segmentation. In these areas, the computational efficiency gained through our pruning method is crucial for deploying models in resource-constrained environments, such as mobile devices and edge computing platforms.
>
> 3. Specific Use Cases
>
> - Image Classification in Privacy-Sensitive Contexts: Given DIMAP’s data-independent nature, it is particularly suited for scenarios where privacy is paramount. For instance, in healthcare imaging, where patient data confidentiality is a priority, DIMAP can prune and optimize models without requiring access to sensitive patient images.
>
> - Object Detection in Real-Time Systems: In real-time surveillance or autonomous driving systems, the reduced model complexity and increased inference speed achieved by DIMAP can be vital. Here, our method can help in maintaining high accuracy while ensuring real-time performance.
>
> - Semantic Segmentation in Mobile Applications: For mobile applications requiring semantic segmentation, such as augmented reality (AR) or navigation aids, DIMAP’s efficiency can enable more complex models to run on limited-resource devices.
>
> 4. Comparison with Existing Approaches
>
> Compared to traditional pruning methods that often rely on data-dependent processes, DIMAP’s data-independent approach offers a significant advantage in terms of privacy and applicability in diverse environments. Furthermore, by focusing on module-level contributions, our method ensures a more targeted and effective pruning process, which we have demonstrated to be more efficient than existing techniques in our experiments with Swin Transformers.
>
> 5. Future Directions
>
> We are exploring several exciting avenues for future research:
>
> - Adapting DIMAP for Other Network Architectures: Expanding the applicability of DIMAP to other types of neural networks, including Convolutional Neural Networks (CNNs) and Recurrent Neural Networks (RNNs), could further broaden its utility.
>
> - Exploring Non-Vision Applications: Investigating the use of DIMAP in non-vision tasks, such as natural language processing or time-series analysis, could reveal additional areas where our method excels.
>
>
> We hope this response adequately addresses your query and highlights the potential of DIMAP in various applications. We are grateful for the opportunity to enhance our manuscript with this discussion and look forward to further feedback.

---

### Meta-Review · Area_Chair_JWCy · 2023-12-08

**Metareview:**

The paper is about the pruning of vision transformers. All the reviewers provide positive ratings for this paper. I think the idea of data-independent-non-uniform pruning looks somewhat novel. Although some reviewers have a concern about the generalization ability of the proposed method into other structures of vision transformers, the response properly addresses these concerns. It would be better if the author could add more experiments on other variants of vision transformers to validate the method further. After comparing the papers in the pool, I decided to accept this paper.

**Justification For Why Not Higher Score:**

N/A

**Justification For Why Not Lower Score:**

N/A

---

### Decision · Program_Chairs · 2024-01-16

Accept (poster)